# Total Colectomy as a Part of Ultra-Radical Surgery for Ovarian Cancer—Short- and Long-Term Outcomes

Sebastian Szubert [1], Artur Skowyra [1], Andrzej Wójtowicz [2], Pawel Pawlowicz [1], Marek Szczepkowski [3], Blazej Nowakowski [4] and Lukasz Wicherek [1,*]

[1]  2nd Department of Obstetrics and Gynecology, Centre of Postgraduate Medical Education, 01-809 Warsaw, Poland; szuberts@o2.pl (S.S.); skowyra.artur@gmail.com (A.S.); ppawlowiczpro@gmail.com (P.P.)

[2]  Department of Artificial Intelligence, Faculty of Mathematics and Computer Science, Adam Mickiewicz University in Poznan, 61-614 Poznan, Poland; andre@amu.edu.pl

[3]  Clinical Department of Colorectal, General and Oncological Surgery, Centre of Postgraduate Medical Education, 01-809 Warsaw, Poland; m.szczepkowski@bielanski.med.pl

[4]  Surgical, Oncology and Endoscopic Gynecology Department, The Greater Poland Center Cancer, 61-866 Poznan, Poland; blanowak@o2.pl

*  Correspondence: mowicher@cyf-kr.edu.pl

**Abstract:** (1) Background: The aim of this study was to assess the outcomes for patients who underwent total colectomy (TC) as a part of surgery for ovarian cancer (OC). (2) Methods: We performed a retrospective analysis of 1636 OC patients. Residual disease (RD) was reported using Sugarbaker's completeness of cytoreduction score. (3) Results: Forty-two patients underwent TC during primary debulking surgery (PDS), and four and ten patients underwent TC during the interval debulking surgery (IDS) and secondary cytoreduction, respectively. The median overall survival (mOS) in OC patients following the PDS was 45.1 months in those with CC-0 (21%) resection, 11.1 months in those with CC-1 (45%) resection and 20.0 months in those with CC-2 (33%) resection ($p = 0.28$). Severe adverse events were reported in 18 patients (43%). In the IDS group, two patients survived more than 2 years after IDS and one patient died after 28.6 months. In the recurrent OC group, the mOS was 6.9 months. Patient age above 65 years was associated with a shortened overall survival (OS) and the presence of adverse events. (4) Conclusions: TC as a part of ultra-radical surgery for advanced OC results in high rates of optimal debulking. However, survival benefits were observed only in patients with no macroscopic disease.

**Keywords:** ovarian cancer surgery; total colectomy; cytoreductive surgery; debulking surgery

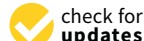



## 1. Introduction

Ovarian cancer (OC) is the leading cause of death from gynecological malignancies in the Western world [1]. Cytoreductive surgery followed by platinum-based chemotherapy is now recommended for the treatment of patients with OC [2,3]. Current research indicates a strong relationship between the amount of residual disease following the initial surgery and patient prognosis. In a meta-analysis by du Bois et al., the authors showed a threefold higher median overall survival (mOS) in patients whose surgery resulted in no gross macroscopic disease when compared to patients with minimal macroscopic residual disease (tumors in diameter < 1 cm) and to patients with nonoptimal debulking (tumors > 1 cm in diameter) [4]. The same research group showed that improvement in surgical skills, and thereafter the widening of the range of surgery, almost doubled the survival rates for OC patients [5].

The main route of OC spread is related to the flow direction of peritoneal fluid circulation, while the seeding of cancer cells is mainly observed in the regions of ascites accumulation, like the pouch of Douglas, the right subphrenic region and the paracolic gutters [6–8]. Colon involvement is observed in more than one-third of OC patients [9,10].

Furthermore, in the majority of cases of advanced OC the colon is involved in multiple segments [11,12]. One of the serious problems in multiple bowel resection in advanced OC is related to anastomotic leakage. Therefore, to avoid the risk of anastomotic leakage, metastases to multiple sites in the colon may be managed with total colon resection. In most cases, it is feasible to widen the range of the surgery and perform an en bloc tumor resection with a modified posterior pelvic exenteration incorporating adnexa, uterus, rectum and pelvic peritoneum, and then extend to a total colectomy (TC) with omentectomy, splenectomy and diaphragmatic and parietal peritonectomy. In a study by Song et al., this type of extensive surgery was reported with very good five-year overall survival [10]. A number of studies have confirmed that OC patients without macroscopic residual disease after primary (PDS) or interval (IDS) debulking surgery have the best prognosis [4,12]. The results of these studies have established the achievement of no ("zero") macroscopic disease as the principal rule for OC surgery. However, there is an evident association between the patient selection and the complete resection rate [13]. Furthermore, the final patient prognosis is not only related to the amount of residual disease. In one recent study, Horowitz et al. have shown that aside from any residual disease the initial tumor burden is directly related to patient outcomes [14]. This suggests that patients who present initially with advanced disease receive less benefit from ultra-radical surgery.

For many years, the surgical management of OC was limited by the surgical skills of the gynecological oncologist. However, it now seems that in the most experienced centers in the world, surgical (technical) skills are no longer the limiting factor; instead, it seems that we have reached the individual patient limits of surgical resection for OC. The most urgent question is whether performing ultra-radical surgery that results in no macroscopic residual disease remains beneficial, on a case-by-case basis, for the patient. Therefore, the aim of this retrospective analysis was to present the perioperative morbidity and the overall survival (OS) of OC patients who had undergone ultra-radical cytoreductive surgery that included the TC.

## 2. Materials and Methods

### 2.1. Study Design

This is the retrospective, observational and descriptive study of patients who underwent TC as a part of the ultra-radical surgery for OC. The data were retrieved from the researchers' current and previous institutions, respectively: The Second Department of Obstetrics and Gynecology, Centre of Postgraduate Medical Education, Warsaw, Poland from 2018–2020, and the Clinical Division of Gynecological Oncology of the Franciszek Lukaszczyk Oncological Center in Bydgoszcz, Poland from 2012–2018. The surgical databases from the hospitals were evaluated both manually and on the basis of the computed search of the assigned procedures of The International Classification of Diseases, 9th Revision, Clinical Modification (ICD-9-CM). We included in the search the procedure numbered 45.8—Total intra-abdominal colectomy.

### 2.2. Patients

Patients were treated in the public health care system on the basis of public health insurance. Patients who were operated on for borderline tumors and nonepithelial tumors were excluded. Patients underwent longitudinal laparotomy extending from the xiphoid process to the pubic bone. Indication for TC was based on significant cancer involvement of the entire colon. The decision whether to perform or abandon surgery was based on the subjective decision of the surgeon. In general, the surgeons' decisions were based on each patient's performance status, the presence of comorbidities and the type of cancer spread.

All surgeries were performed by one accredited gynecological oncologist (L.W.). In all cases, preoperative bowel preparation with a mechanical bowel and a preoperative enema was performed. All patients received an intravenous antibiotic prophylaxis composed of first-generation cephalosporin, metronidazole and gentamicin. Most patients who underwent extensive surgery received postoperative parenteral nutrition. The administration

of transfusions of red blood cell concentrates (RCC) depended on the patient's clinical performance; however, most patients with postoperative hemoglobin concentrations below 8 g/dL received RCC.

For all the removed tumors, final histopathological diagnosis was obtained, and the tumors were classified according to the World Health Organization (WHO) guidelines. The stage of the disease was then assessed according to the 2014 FIGO classification. Cases treated prior to 2014 were reclassified using the 2014 FIGO staging.

Except for cases of early mortality, all patients received first-line chemotherapy consisting of intravenous carboplatin and paclitaxel.

The study group was divided into two subgroups: (1) patients who underwent TC during primary debulking surgery (PDS); (2) patients who had TC performed after chemotherapy for OC. We compared patient survival rates according to the completeness of cytoreductive (CC) surgery (using Sugarbaker's completeness of cytoreduction score [15]). In summary, no macroscopic residual disease was scored as CC = 0, while CC-1 scored as nodules below 2.5 mm after surgery; CC-2 as nodules between 2.5 mm and 2.5 cm; CC-3 as nodules >2.5 cm.

### 2.3. Outcomes

Information on any patients who died was retrieved from the database of the National Health System of Poland (March 2021). We presented both the short-term (postoperative adverse events) and long-term outcome (overall survival (OS)) of the surgical procedure. Postoperative adverse events were ranked according to the Clavien–Dindo classification [16]. We included only grade 3 or more adverse events. Perioperative mortality was defined as death due to any reason within 30 postoperative days. The OS was defined as the length of time from the date of cytoreductive surgery with TC to the time of death.

### 2.4. Explanatory Variable

The first explanatory (independent) variable in our study was the surgical procedure of TC during cytoreductive surgery for OC. To evaluate the effect of TC on patient survival in the group of patients treated during PDS, we investigated the impact of the following confounders on patient survival: the presence of adverse events (grade 3 or more according to Clavien–Dindo classification [16]), diaphragmatic stripping, splenectomy, liver metastasectomy, residual disease (CC-0 and CC-1 vs CC-2 according to Sugarbaker's completeness of cytoreduction score [15]), age (below and above 65), body mass index (BMI; below and above 25) and preoperative albumin level (below and above 30 g/L).

The second explanatory variable in our study was the presence of surgery-related adverse events. We investigated the association between the adverse event occurrence and the following variables: diaphragmatic stripping, splenectomy, liver metastasectomy, lymphadenectomy, residual disease (CC-2 according to Sugarbaker's completeness of cytoreduction score [15]), age (below and above 65), body mass index (BMI; below and above 25), preoperative albumin level (below and above 30 g/L) and previous chemotherapy.

### 2.5. Statistical Analysis

Comparison of the groups according to the CC score was conducted using the Fisher's exact test and the Kruskal–Wallis test.

Survival analyses were conducted using the Kaplan–Meier survival curves and the differences in patient survival were compared using the log-rank test. The multivariate survival analysis was conducted using Cox proportional-hazards regression with the stepwise method of variable entry. All of the confounders listed in Section 2.4 were used for the model development. The stepwise method indicates that significant variables are entered into the model sequentially. After entering the variable is rechecked, nonsignificant variables are removed.

The unadjusted and adjusted odds ratio (OR) analysis was performed to evaluate the impact of surgical procedures and patient characteristics on the presence of adverse events.

Statistical analysis was carried out using: MedCalc 11.4.2.0, MedCalc Software Ltd., Ostend, Belgium; GraphPad InStat 3.06, GraphPad Software, San Diego, CA, USA; and R v4.0.2 software, R Core Team, R Foundation for Statistical Computing, Vienna, Austria.

## 3. Results

### 3.1. Patient Characteristics

We identified 83 patients who had been surgically treated for OC in our present place of work, the Second Department of Obstetrics and Gynecology, Centre of Postgraduate Medical Education, Warsaw, Poland within the analyzed period. Of these, six (7%) patients had undergone TC. In our previous workplace, the Clinical Division of Gynecological Oncology of the Franciszek Lukaszczyk Oncological Center in Bydgoszcz, of 1553 patients who were operated on for OC, 50 (3%) had undergone TC. Therefore, of an overall total of 1636 OC patients who had been treated, TC had been performed in 56 (3%) patients; thus, our study group was comprised of those 56 patients. The median patient age was 58 years (range 26–78). The median follow-up period was 38 months.

All the patients underwent TC with a modified posterior pelvic exenteration (i.e., an en bloc removal of the uterus or vaginal vault in the case of previous hysterectomy, rectum, bilateral adnexa and pelvic peritoneum). To avoid the risk related with anastomotic leakage, a final ileostomy was made in all cases. Additionally, a total omentectomy was performed. All the patients underwent small bowel resection. The range of the small bowel resection was dependent on the tumor involvement; however, we did not perform any resection resulting in the shortening of the small bowel to more than 150 cm. Other procedures, like diaphragmatic peritonectomy, splenectomy or resection of liver metastases were performed when necessary, depending on the degree of tumor infiltration, in order to remove all macroscopic lesions. A lymphadenectomy was always performed in those cases where enlarged or suspicious lymph nodes were found. In cases where the lymph nodes were unchanged, the primary surgeon decided whether to perform a lymphadenectomy. The examples of surgical specimens are presented in Figure 1.

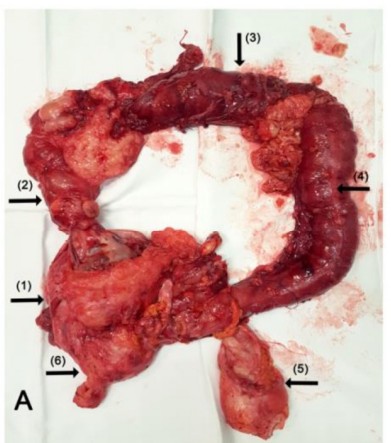 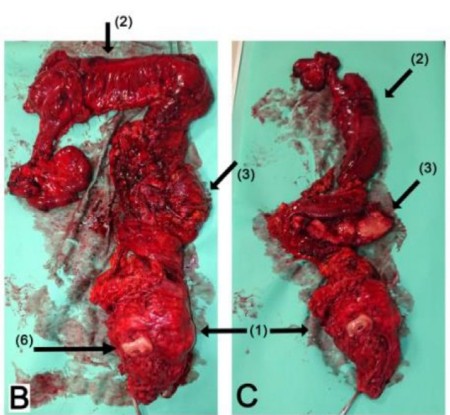 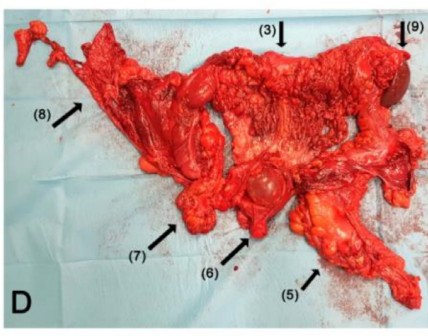

**Figure 1.** TC during debulking surgery for advanced OC. The photography of en bloc resected surgical specimen of TC: (**A**) the specimen after primary debulking surgery due to mucinous OC; (**B**,**C**) the specimen after interval debulking surgery due to serous OC; (**C**) the reverse view of transverse colon and "omental cake"; (**D**) the specimen after primary debulking surgery due to high-grade serous carcinoma. Arrows: (1) the ovarian tumor with uterus and pelvic peritoneum; (2) ascending colon; (3) transverse colon; (4) descending colon; (5) rectum; (6) uterine cervix; (7) distal ileum; (8) distal ileum; diaphragmatic and parietal peritoneum; (9) spleen.

The median duration of surgery and the median hospital stay were 285 min (125–530 min) and 20 days (7–116 days), respectively. In the whole study group, 24 patients (43%) experienced severe adverse events. The most common adverse event was wound infection and occurred in 11 (20%) of the patients. Therefore, 23% of our patients experienced severe surgical complications other than wound infections. The median surgery–chemotherapy

interval was 31 days (range 9–89 day). However, six patients (11%) did not receive adjuvant chemotherapy due to death or significant morbidity. The median patient survival in the whole group was 20.1 months (range 0.9–72.7).

In the whole study group, we found no association between the occurrence of surgical-related adverse events and the analyzed factors, both in the univariate and multivariate analysis (Table 1).

**Table 1.** Unadjusted and adjusted odds ratios (OS) for variables included in the logistical regression model for the occurrence of any adverse event following TC during cytoreductive treatment of OC.

| Variable | Unadjusted OR (95% CI) | *p*-Value | Adjusted OR (95% CI) | *p*-Value |
|---|---|---|---|---|
| Diaphragmatic stripping | 0.75 (0.34–1.57) | 0.451 | 1.19 (0.11–15.35) | 0.881 |
| Splenectomy | 0.73 (0.36–1.46) | 0.386 | 1.25 (0.09–14.35) | 0.854 |
| Liver metastasectomy | 4.69 (0.51–136.61) | 0.232 | 21.40 (1.42–1467.01) | 0.060 |
| Residual disease CC-2 | 0.75 (0.24–2.15) | 0.594 | 1.14 (0.30–4.31) | 0.842 |
| Lymphadenectomy | 0.59 (0.29–1.15) | 0.133 | 0.55 (0.12–2.31) | 0.427 |
| Previous chemotherapy | 0.37 (0.08–1.29) | 0.147 | 0.38 (0.05–1.92) | 0.271 |
| Age > 65 | 0.33 (0.07–1.11) | 0.099 | 0.42 (0.07–1.78) | 0.263 |
| BMI > 25 | 0.49 (0.19–1.13) | 0.106 | 0.29 (0.06–1.14) | 0.089 |
| Albumin level < 30 g/L | 0.66 (0.30–1.41) | 0.295 | 1.84 (0.41–8.71) | 0.418 |

95% CI—95% confidence interval; residual disease refers to Sugarbaker's completeness of cytoreduction score [15], where patients with CC-2 had residual nodules between 2.5 mm and 2.5 cm in diameter.

### 3.2. Primary Debulking Surgery

We identified 42 patients who underwent a TC as part of ultra-radical surgery for OC during the PDS. In addition, 22 (52%) patients underwent diaphragmatic surgery, 26 (62%) patients had splenectomy, 3 (7%) patients had a resection of liver metastases, 22 (52%) patients had both a pelvic and paraaortic lymphadenectomy and 4 (10%) patients only had a pelvic lymphadenectomy. All the patients were classified as stage IIIC according to FIGO staging except for one case of IIIB, three cases of IVA and three cases of IVB. The histopathological types of the tumors were as follows: high grade serous OC—29 (69%); clear cell cancer—2 (5%); endometrioid OC—2 (5%); undifferentiated OC—4 (9%); mucinous OC—5 (12%); carcinosarcoma of the ovary—1 (2%).

In 9 cases the surgery resulted in no gross residual disease (CC = 0, 21%), 19 (45%) patients had residual disease defined as CC-1, and 14 (33%) patients had CC-2. There were no patients with CC-3 resection. Therefore, at least two-thirds of the patients had optimal (tumors less than 1 cm) debulking. Across the whole group, the median patient OS was 24.2 months (range 0.9–72.7). Patients with CC-0 resection had a median overall survival (mOS) of 45.1 months (range 3.8–56.7), while the patients with CC-1 and CC-2 resection had an mOS of 11.1 months (0.9–72.7) and 20.0 months (0.4–48.5), respectively (*p* = 0.28). The survival curves corresponding with the degree of the residual disease following the TC are presented in Figure 2.

In the multivariate, adjusted, survival analysis, we found that the presence of severe adverse events (Hazard Ratio (HR) = 1.66; 95% CI = 1.32–20.1) and an age above 65 years (HR = 2.21; 95% CI = 1.47–56.6) were independently related with shortened overall survival. Diaphragmatic stripping, splenectomy, residual disease, BMI, resection of liver metastases and preoperative albumin levels were not related with the patient OS among those patients who had a TC during the PDS for OC.

Severe adverse events (Grade 3 or more in the Clavien–Dindo classification [16]) were reported in 18 patients (43%). Grade 4 surgical complications were observed in five (12%) patients. Two patients (4.8%) died in the perioperative (Grade 5 according to Clavien–Dindo [16]) period. The perioperative morbidity is summarized in Table 2.

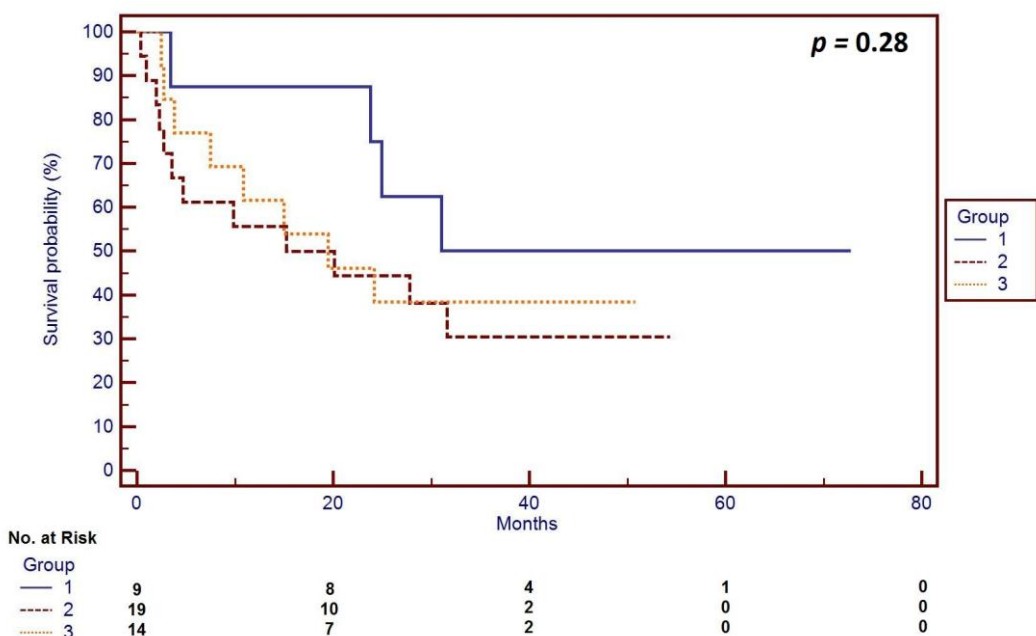

**Figure 2.** Survival curves corresponding with the degree of residual disease following TC during primary debulking surgery for OC according to the completeness of cytoreductive (CC) surgery (using Sugarbaker's completeness of cytoreduction score [15]): Group 1—patients (*n* = 9) with no residual disease (CC = 0), median overall survival (mOS) of 45.1 months (range 3.8–56.7); Group 2—patients (*n* = 19) with nodules below 2.5 mm after surgery (CC-1), mOS of 11.1 months (0.9–72.7); Group 3—patients (*n* = 14) with nodules between 2.5 mm and 2.5 cm (CC = 2), mOS of 20.0 months (0.4–48.5); *p* = 0.26.

**Table 2.** The analysis of adverse events and perioperative outcomes among patients treated with total colectomy for ovarian cancer.

| Adverse Events. | CC = 0 *n* = 9 (21%) | CC = 1 *n* = 19 (45%) | CC = 2 *n* = 14 (33%) | *p*-Value |
|---|---|---|---|---|
| Patients affected by severe adverse events | 4 (44%) | 9 (50%) | 5 (38%) | *p* = 0.91 |
| Perioperative mortality (within 30 days) | 1 (11%) | 1 (5%) | 0 (0%) | *p* = 0.69 |
| Wound infection | 2 (22%) | 3 (16%) | 2 (14%) | *p* = 1.0 |
| Intra-abdomen abscess | 1 (11%) | 1 (5%) | 0 (0%) | *p* = 0.69 |
| Ileus requiring surgery | 1 (11%) | 0 (0%) | 0 (0%) | *p* = 0.22 |
| Cerebral infraction | 0 (0%) | 1 (5%) | 1 (7%) | *p* = 0.99 |
| Cardiac infraction | 0 (0%) | 1 (5%) | 2 (14%) | *p* = 0.58 |
| Intestinal fistula | 0 (0%) | 2 (11%) | 0 (0%) | *p* = 0.69 |
| Repeated surgery | 1 (11%) | 3 (16%) | 1 (7%) | *p* = 0.83 |
| Cholecystitis with gallbladder perforation | 0 (0%) | 0 (0%) | 1 (7%) | *p* = 0.55 |
| Median red blood cells transfusion rate | 2 (0–10) | 2 (0–12) | 2 (0–8) | *p* = 0.59 |

CC score—completeness of cytoreductive (CC) surgery according to Sugarbaker [15]. Severe adverse events were defined as at least a Grade 3 complication according to the Clavien–Dindo classification [16].

### 3.3. Surgery following Chemotherapy

We identified 14 patients who had undergone TC as a part of their surgical treatment for OC following initial chemotherapy. The histopathological types of the tumor were as follows: 10 high-grade serous OC, 1 endometrioid carcinoma and 3 undifferentiated OC. Among these 14 patients, 4 had a TC during the interval debulking surgery (IDS) following two, three and six courses of platinum-based neoadjuvant chemotherapy (NACT), respectively. Two of these patients survived more than 2 years after the IDS, and one patient died 28.6 months after the IDS. Of the four patients treated with IDS, two (50%) patients had wound infections. All the patients received the rest of their chemotherapy in the planned time.

Ten patients were treated surgically for the recurrence of persistent OC. Within this subgroup, the mOS was 6.9 months (range 3.2–39.0). However, two (20%) patients were in good health for nearly 3 years. Four (40%) patients had severe (at least Grade 3) adverse events. Detailed patient characteristics are presented in Table 3.

**Table 3.** Characteristics of the OC patients who underwent TC as a part of surgical treatment during interval debulking surgery or secondary surgery due to cancer recurrence.

| | Previous Treatment | Survival after Surgery (Months) | CC Score | Severe Adverse Events | Diaphragmatic Resection | Splenectomy | Liver Metastases Resection |
|---|---|---|---|---|---|---|---|
| colspan | Interval debulking surgery (IDS) after neoadjuvant chemotherapy (NACT) due to ovarian cancer (OC) | | | | | | |
| 1 | IDS after 6 courses of NACT | 28.6 | 0 | Wound infection | No | No | No |
| 2 | IDS after 2 courses of NACT | 34.6 (alive) | 0 | NR | Yes | No | Yes |
| 3 | IDS after 3 courses of NACT | 24 (alive) | 1 | NR | No | Yes | No |
| 4 | IDS after 3 courses of NACT | 1 (alive) | 1 | Wound infection | No | No | No |
| colspan | Secondary cytoreduction after OC recurrence | | | | | | |
| 1 | OC first recurrence, after 3 years | 5.3 | 1 | Abdominal abscess | Yes | Yes | No |
| 2 | OC first recurrence, after 4 years | 3.5 | 1 | NR | No | No | No |
| 3 | OC first recurrence, after 3 years | 5.7 | 2 | NR | No | No | No |
| 4 | Persistent OC following 2 lines of chemotherapy. Palliative surgery due to ileus | 3.2 | 1 | Fatal cerebral infraction | Yes | Yes | No |
| 5 | Second recurrence after 3 lines of chemotherapy | 6.9 | 1 | NR | Yes | Yes | No |
| 6 | OC first recurrence after 2 years, then 3 courses of chemotherapy | 6.4 | 1 | Wound infection | Yes | Yes | No |
| 7 | OC progression after 6 months of stable disease following bevacizumab | 11.1 | 1 | NR | Yes | Yes | No |
| 8 | OC first recurrence after 2 years | 39.0 (alive) | 1 | Pancreatic fistula, reoperation, abdominal wall infection | Yes | Yes | Yes |
| 9 | OC first recurrence after 4 years, then 3 cycles of chemotherapy | 34.6 (alive) | 1 | NR | No | No | No |
| 10 | OC first recurrence after 1 year | 7.1 (alive) | 0 | NR | Yes | Yes | No |

CC score—completeness of cytoreductive (CC) surgery according to Sugarbaker [15]. Severe adverse events were defined as at least a Grade 3 complication according to the Clavien–Dindo classification [16].

## 4. Discussion

We have reported the short- and long-term outcomes for patients who underwent ultra-radical surgery composed of a TC for advanced OC. This type of surgery is very rarely performed on OC patients, and in our study it was performed in only 3% of patients. Our study describes the outcomes for 56 patients, and to the best of our knowledge this is the largest study reporting TC as a part of the surgical management of OC. Other studies have reported on fewer patients: Song et al. [10] reported on 22 patients; Oseledczyk et al. [17] on 11 patients; Bacalbasa et al. [18] on 17 patients; Turnbull et al. [19] on 10 patients; Walter et al. [20] on 9 patients. Further studies, such as Silver et al. [21], Hoffman et al. [22] and [9] and Cascales Campos et al. [23] have each reported only a few patients. In addition, these studies included heterogenous groups of patients, reporting on cases that had TC during PDS, IDS and secondary surgery during cancer recurrence, as well as surgery during palliative indications.

In our study we observed that OC patients who had the PDS with no gross residual disease (CC-0) had an mOS of 45.1 months. Although the difference between the patients with CC-0 resection compared to patients with CC-1 and CC-2 was not significant, the trend toward the better survival of patients with CC-0 resection was observed. We have traditionally used Sugarbaker's completeness of cytoreduction scale to describe the amount of residual disease. In this context, CC-1 is defined as the tumor nodules below 2.5 mm, referring to a very low burden of residual disease. On the one hand, the difference between CC-0 and CC-1 resection is not substantial, but on the other hand, these two groups may include biologically different tumors [24]. Furthermore, the lack of statistical significance between the analyzed subgroups may be attributed to the less apparent benefit from the radical surgery that is observed in more advanced disease stages [14]. Finally, the lack of differences in the patient survival rates may be due to the low number of patients within our subgroups.

The survival of the patient treated during the PDS in our study seems to be comparable with the results achieved by Oseledchyk et al. [17], who reported the OS data for five OC patients treated with TC during the PDS. The OS of their patients ranged between 1.78 and 38.3 months [17]. Song et al. have reported a 74.4% five-year OS [10]. In this context, the survival reported by Song et al. is much better than that observed in our study and in the study by Oseledchyk et al. [17]. However, we had a higher rate of splenectomy (57%) compared to Song et al., and splenectomy is typically associated with severe morbidity, especially when associated with bowel surgery [10,25]. Furthermore, the study by Song et al. included patients of Asian ethnicity, while our study included only Caucasians, and this latter group has a worse prognosis than the former [26]. More than two-thirds of our patients had surgery that resulted in optimal debulking (i.e., tumors less than 1 cm). These results are very similar to those of Oseledchyk et al. [17], who achieved optimal debulking in 64% of cases. Similarly, the rate of complete (i.e., no gross residual disease) resection was 21% in our study and 18% in the study by Oseledchyk et al. [17]. On the other hand, in the study by Song et al. [10], the authors reported no gross residual disease in 45.5% of patients, while Bacalbasa et al. reported the complete (CC-0) cytoreduction in all their cases [18]. Therefore, the better survival rate reported by Song et al. [10] may also be attributed to their higher rate of CC-0 resection.

In the case of the management of the newly diagnosed advanced OC, the key question is whether to proceed with the ultra-radical PDS with TC or to begin with neoadjuvant chemotherapy and then perform the IDS. There is no data that clearly answers this question. The longest survival of OC patients has only been achieved when the upfront surgery resulted in no gross residual disease (CC = 0) [4,12]. The first two studies comparing PDS and IDS performed by Vergote et al. [27] and the CHORUS trial [28] showed similar long-term outcomes for both groups; nevertheless, the patients treated with IDS had significantly lower rates of severe adverse events. However, the patients from both groups also had poor survival. More recent studies performed in Italy (SCORPION trial [29]) and Japan by Onda et al. [30] have shown similar results when the PDS was compared with the IDS; however, the patient prognosis in these studies was better than those reported by Vergote et al. [27] and in the CHORUS trial [28]. The SCORPION trial [29] and the Japanese [30] study precisely compare the PDS and IDS groups; however, these trials did not contain sufficient data to evaluate the role of colectomy in OC surgery. In our study, we have shown that only 3% of patients treated due to OC underwent TC. Similar results were provided by Oseledchuk et al., who performed TC in 3% of surgeries for OC [17]. The absence of trials evaluating the role of TC in OC surgery suggests that this procedure is infrequently performed. Our trial used subjective criteria to qualify patients for TC that were based on personal experience. We have observed that advanced patient age (above 65 years of age) and the presence of adverse events were related with shortened survival. However, we did not identify those factors associated with surgery-related adverse events.

Advanced age is a known risk factor for worse prognosis in OC [31]. The OC patients above 65 years of age are more likely to have incomplete surgery and more residual disease following the surgery [32]. In general, the advanced age OC patients experience similar or fewer rates of surgically related adverse events because the surgery in this group of patients is less radical [33,34] However, when radical surgery is applied, advanced patient age is related with a higher rate of adverse events [35]. Therefore, in the case of an advanced age OC patient, special care must be taken when TC is considered. We believe that only prospective trials that precisely describe the extent of the surgery can help in solving the problem and define where the individual patient limits of surgical interventions in advanced OC lie. Considering recent studies showing varying prognoses depending on the type of peritoneal spread in cases of high grade serous OC, a precise assessment of the colectomy in the upfront debulking surgery should include the molecular heterogeneity of the cancer disease [24,36,37].

Our study also included patients who had ultra-radical surgery with TC following chemotherapy (either neoadjuvant or secondary). In these patients, the surgery was treated

as a salvage therapy alternative to palliation. Although this group of patients was small and very heterogenous, we observed promising survival rates in those patients who underwent ultra-radical surgery following the first three or six cycles of chemotherapy. This group of women represented patients with limited platinum sensitivity. However, in those cases of patients with a limited response to neoadjuvant chemotherapy, where upper abdomen involvement is still present, we consider TC with ultra-radical surgery as a reasonable course of management when complete or optimal debulking is possible. In the cases of those treated with TC during OC recurrence, we reported poor overall survival. Oseledchyk et al. have reported four cases of TC due to OC recurrence. Patient survival following surgery for the recurrence ranged between 7.76 and 10.62 months [17]. These results are similar to those in our study. In the DESKTOP III trial, du Bois et al. [38] showed that OC patients with a first relapse and positive AGO score benefit from secondary cytoreduction when a complete resection was achieved. The authors found that in the group of patients who were treated with chemotherapy alone, the progression-free survival was 14 months. In our study we did not follow AGO score criteria when qualifying the patients for the secondary debulking surgery; however, most of our patients were platinum sensitive. Only two (20%) of our patients who were treated due to OC recurrence seemed to benefit from the surgery. These patients were among the last to be operated on in our study group; therefore, it is possible that their increased survival is associated with the improvement of surgical skills. However, further studies are needed to clarify the indications and limits for secondary cytoreduction.

In our study, severe surgical complications (Grade 3 or more) were observed in 43% of the patients. The rate of severe adverse events in the group of OC patients who had the IDS or secondary debulking was similar. The most common surgical complication, accounting for nearly half of the adverse events, was wound infection. It seems that extensive surgery and a lower albumin level, when related with neoplastic disease, results in aberrant wound healing that is related with secondary infection. However, wound infection in our patients was well managed with Vacuum-Assisted Closure (VAC) of the wound with early surgical resuturing. The need for wound resuturing is classified as a Grade 3 wound infection, and so the rate of severe adverse events in our study is high. Therefore, taking into account the extent of the surgery and the rate of other and more severe adverse events (23% in our study), it was in our opinion acceptable. However, 11% of the patients did not receive the adjuvant chemotherapy due to death or significant morbidity. In the study by Oseledchyk et al., severe Grade 3–4 postoperative complications occurred in three (27%) patients, and one (9%) patient died within the perioperative period [17]. Song et al. have reported adverse events in 31.8% of patients, while Bacalbasa et al. have described 3 (17,6%) severe adverse events among 17 OC patients treated with TC [10,18]. In this context, the rate of surgical adverse events after TC during the primary surgery seems to be close to that as for those patients treated with an aggressive surgery during the PDS, and higher than for those patients who underwent NACT [29,30]. To lower the incidence of surgical complications, all our patients underwent terminal ileostomy. However, other authors have also shown that restoration of bowel continuity after TC may be a safe option also for OC patients [10,17,18].

## 5. Conclusions

Our results and the review of the literature suggest that TC, as a part of ultra-radical surgery, for advanced OC results in a high rate of optimal debulking and is associated with a high rate of severe adverse events. However, the morbidity is predominantly related with postoperative wound infection that is well managed. The rate of other severe adverse events is also acceptable. Despite the high rate of optimal cytoreduction, the survival benefits from ultra-radical surgery were observed only in patients with no macroscopic disease. Therefore, it seems that TC in OC is reasonable only when complete resection is achieved.

**Author Contributions:** Conceptualization: S.S., L.W., A.W., A.S. and B.N.; resources: L.W., S.S., M.S., P.P. and B.N.; formal analysis: A.W., S.S., M.S., A.S., P.P. and L.W.; writing—original draft: S.S.,

A.S. and P.P.; writing—review and editing: L.W., M.S. and B.N.; supervision L.W., A.W. and M.S.; correspondence: L.W. All authors have read and agreed to the published version of the manuscript.

**Funding:** This research received no external funding.

**Institutional Review Board Statement:** The study was approved by the Centre of Postgraduate Medical Education Ethics Committee (8/PB/2020; 5 January 2020).

**Informed Consent Statement:** Informed consent was obtained from all subjects involved in the study.

**Data Availability Statement:** Data available on request from the authors.

**Acknowledgments:** We would like to thank Robert Garret for his assistance with the manuscript.

**Conflicts of Interest:** The authors declare no conflict of interest.

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
