# Peer review of "Total Colectomy as a Part of Ultra-Radical Surgery for Ovarian Cancer—Short- and Long-Term Outcomes"

_curroncol, doi:10.3390/curroncol28050358_

Round 1

Reviewer 1 Report

Significant improvements were made to the methods and discussion sections. I think the outcomes were better described. I think I would highlight the fact that OS was excellent in patients undergoing total colectomy to achieve R0, but OS was not very good in patients with macroscopic residual disease. I think this shows that TC is reasonable if it allows you to achieve R0, but otherwise the survival outcomes do not justify the increased morbidity.

Author Response

Dear Reviewer,

We would like to thank you very much for your efforts in reviewing our manuscript. We have modified the manuscript according to your suggestions. Thank you very much.

Reviewer 2 Report

Dear Authors,

thank you very much for the opportunity to review this paper.

This is a unique case series of a very heavy surgery for a mostlikely uncurable disease such as ovarian cancer. The topic is of interest and I do not have any major concerns, since this a purely descriptive paper and moreover, you have stated in the abstrat an honest conclusion regarding the high rate of comorbidites without an improvement in survival.

So far also the analysis, even though with a small sample size, are correct in the method.

I only suggest to cut the introduction (too verbose).

Best regards 

Author Response

Dear Reviewer,

We would like to thank you very much for your efforts in reviewing our manuscript. We have modified the manuscript according to your suggestions. Thank you very much.

This manuscript is a resubmission of an earlier submission. The following is a list of the peer review reports and author responses from that submission.

Round 1

Reviewer 1 Report

The authors aim to describe the outcome of women with advanced ovarian cancer subjected to colectomy at time of cytoreductive surgery. The research question has merit in its clinical significance and is of interest. However, there are major concerns related to the methodology used to answer the research question. And for this reason, I would recommend the authors to rewrite the paper with suggestions as to below.

First the authors need to define what their research question is, it is not clear:

To describe postoperative outcomes after colectomy?

To describe oncologic outcomes after colectomy?

Is there a comparison group (is this a cohort study to compare outcome in comparison to those who have not been subjected to colectomy? Or a comparison between timing of surgery and outcomes after colectomy)

Or is this a descriptive observational study describing the odds of postoperative complications after colectomy where no adjustment for other variables have been made?

Would suggest excluding patients subjected to surgery in the recurrent situation to have a more homogenous material.

ABSTRACT

Must be rewritten after the below has been altered

INTRODUCTION

Inform the reader that association between complete resection is dependent on patient selection (see and reference Falconer et al Gynecol Oncol 2020)

Define in the last paragraph the aim of the study, i.e., describe outcome in women who have been subjected to colectomy at time of cytoreductive surgery??

METHODS

 Inform the reader of:

  • the setting: private hospitals, public health care system, insurance based etc.
  • if the study has ethical approval.
  • Is there a cancer registry in the country of the authors?
  • If so, to which extent does the material account for the population?
  • Inform the reader of the study design, is this a cohort study or an observational descriptive study 
  • How were patients selected for surgery, was there an algorithm?
  • How were the variables extracted and from what database?

Insert a heading: Patients (and describe your population)

Insert a heading: Outcome (and define the outcome of the study). If the outcome is not oncologic, however postoperative adverse event. Describe and define adverse events in detail, which classification was used, and in which time-period after surgery?

Insert a heading: Predictor/Explanatory variables (and describe why each variable were) chosen as confounders and which variable is the variable you are studying (colectomy?)

A cox regression controlling for the predictor variables chosen are needed in addition to the descriptive K-M curves, no conclusion could otherwise be made.

If the primary outcome is adverse events, define the predictor variable of interest and control for other variables associated with adverse events in the multivariable regression model.

Insert a heading: Statistical method and describe how the cox/multivariable regression has been performed.

Exclude Figure 1 from the Methods section and a large part of the methods section describing the surgical procedure should be moved to the results section.

RESULTS

Provide a flow chart of how the final patient material was selected.

Provide a multivariable regression model (or cox) adjusting for confounders.

DISCUSSION

Pending new results according to the above.

Author Response

Dear Reviewer,

We would like to thank you very much for your efforts in reviewing our manuscript. We have addressed all of the issues raised in your remarks and we have revised our manuscript accordingly. Thanks to your remarks, we believe that our manuscript is improved. Please find our responses below:

Reviewer 1:

Question:

The authors aim to describe the outcome of women with advanced ovarian cancer subjected to colectomy at time of cytoreductive surgery. The research question has merit in its clinical significance and is of interest. However, there are major concerns related to the methodology used to answer the research question. And for this reason, I would recommend the authors to rewrite the paper with suggestions as to below.

First the authors need to define what their research question is, it is not clear:

To describe postoperative outcomes after colectomy?

To describe oncologic outcomes after colectomy?

Answer: The aim of our study was to present both short-term (postoperative period, the rate of adverse events) and long-term (overall survival) of ovarian cancer patients who had total colectomy. We have rewritten the aim of our study.

Question: Is there a comparison group (is this a cohort study to compare outcome in comparison to those who have not been subjected to colectomy? Or a comparison between timing of surgery and outcomes after colectomy). Or is this a descriptive observational study describing the odds of postoperative complications after colectomy where no adjustment for other variables have been made?

Answer: Our study does not have the control group. You are right, that the best way to present the outcomes after total colectomy would be the comparision of the results with the control group composed of patients who had no surgery or had non-optimal surgery and were sent for chemotherapy. However, to resolve the bias related with patients selection (patient comorbidities, heath care organization) such study should be performed as a prospective, randomized trial. However, total colectomy is very rare procedure, therefore, planning of this kind of trial is impossible. We also believe, that the artificial incorporation of a control group in the retrospective analysis will result in significant bias. Therefore, we decided to present the results of the single cohort.

Question: Would suggest excluding patients subjected to surgery in the recurrent situation to have a more homogenous material.

Dear Reviewer, you are right, that excluding of patients who had surgery after cancer recurrence will result in more homoegenous group. However, we would like to present our results. As we noted before, total colectomy is very rare procedure, especially in the case of recurrent OC. Nowadays, it is hard to present the results of 10 patients in a single study, therefore, we would like to present the outcomes of these patients in this study. However, if you decide in the second round of the revision, that our study will benefit from excluding the cases after the recurrence, we would exclude these patients.

Question: ABSTRACT. Must be rewritten after the below has been altered

 Answer: we have added new results to the abstract

INTRODUCTION

Question: Inform the reader that association between complete resection is dependent on patient selection (see and reference Falconer et al GynecolOncol 2020)

Answer : we have added the information about the association between patient selection and complete resection rate

Question: Define in the last paragraph the aim of the study, i.e., describe outcome in women who have been subjected to colectomy at time of cytoreductive surgery??

We have modified the last paragraph of the introduction section.

METHODS

Question: Inform the reader of:

  • Question: the setting: private hospitals, public health care system, insurance based etc.

Answer: corrected

  • Question: if the study has ethical approval.

Answer: We have highlighten the data concering ethical approval.

Question:

  • Is there a cancer registry in the country of the authors?
  • If so, to which extent does the material account for the population?

Answer. We have the national cancer registry in our country. There are about 3500 ovarian cancer cases a year.

  • Question: Inform the reader of the study design, is this a cohort study or an observational descriptive study 

Answer: we have provided the data

  • Question: How were patients selected for surgery, was there an algorithm?

There was no algorithm for patient selection. We have added the information.

  • Question: How were the variables extracted and from what database?

Answer: In general, after the surgery, all of the surgeries have assigned procedures of ICD9 classification. We performed the search of the 45.8 procedure (total intra-abdominal colectomy). However, all of the surgeries were rechecked manually. We have added the data to the manuscript.

Question: Insert a heading: Patients (and describe your population)

Answer: corrected

Question: Insert a heading: Outcome (and define the outcome of the study). If the outcome is not oncologic, however postoperative adverse event. Describe and define adverse events in detail, which classification was used, and in which time-period after surgery?

Answer: We have provided the data about adverse events. We follow the classification by Clavien and Dindo, therefore, we provided the reference for the classification where all of the adverse events were defined. 

Question:

Insert a heading: Predictor/Explanatory variables (and describe why each variable were) chosen as confounders and which variable is the variable you are studying (colectomy?)

A cox regression controlling for the predictor variables chosen are needed in addition to the descriptive K-M curves, no conclusion could otherwise be made.

If the primary outcome is adverse events, define the predictor variable of interest and control for other variables associated with adverse events in the multivariable regression model.

Insert a heading: Statistical method and describe how the cox/multivariable regression has been performed.

Answer: we have corrected the manuscript accordingly. We have developed both the multiple regression model and the Cox regression, and we have presented the results

Question : Exclude Figure 1 from the Methods section and a large part of the methods section describing the surgical procedure should be moved to the results section.

Answer : We have modified the manuscript accordingly.

Question : RESULTS

Provide a flow chart of how the final patient material was selected.

Answer : we have provided the flow chart

Question: Provide a multivariable regression model (or cox) adjusting for confounders.

Answer : we have provided the multiple regression model.

Question:

DISCUSSION

Pending new results according to the above.

Answer: we have added new information concerning new results.

Reviewer 2 Report

This study describes oncologic outcomes and surgical morbidity among patients undergoing total colectomy for ovarian cancer debulking. This is an interesting and timely question, as the use of radical surgery to achieve maximal cytoreduction at the time of primary debulking in ovarian cancer has been questioned by several recent studies (i.e. SCORPION trial, Chorus trial, Vergote, etc.). As the authors point out, rare procedures such as colectomy are difficult to evaluate in randomized  trials, so a retrospective review is appropriate for this question.

Significance: This manuscript is significant, since it is the largest series of patients undergoing total colectomy for ovarian cancer debulking. However, I think some of their conclusions are overstated since there is no comparison group in this descriptive study.

Quality of presentation: The article is written appropriately, and the authors describe the questions and conclusions well. There are some areas where the English level is not that of a native speaker, but overall it is well-written.

Scientific Soundness: The study design is appropriate for the question. This is a small study, but the analyses were performed appropriately. The data is robust enough to draw some conclusions regarding the feasibility of TC in OC debulking. I think some additional analyses could also be considered.

Interest to readers: This should be of interest to readers given the recent questions surrounding the necessity of performing high morbidity procedures for upfront debulking (vs. neoadjuvant chemotherapy with a lower complexity interval debulking surgery).

Overall merit: I think there is an overall benefit to publishing this work, though some additional analyses would improve the merit of this manuscript.

English level: The English is appropriate and understandable, though I think some minor edits would improve the readability of the manuscript.

Comments:

Line 137-139: I’m not sure Sugarbaker’s completeness of cytoreduction score is the best scale to use for this manuscript. I think using residual disease (RD) 0 vs. </= 1 cm vs. > 1 cm would be a more clinically relevant surgical outcome. Additionally, the difference between CC0 and CC1 is very minimal, so that may explain why there is no difference in outcome between these groups. Using the more clinically relevant outcomes of RD0, optimal debulking, and suboptimal debulking could yield different results.

Line 156-165: The authors could report some additional clinical variables including nutrition status (albumin), estimated blood loss (EBL), disease stage, presence of ascites, AGO score, or other factors to more completely describe this group of patients. Could also do some additional analyses to determine whether specific patient characteristics such as nutrition status, age, length of surgery, EBL, ascites, etc. is associated with survival or morbidity in patients undergoing TC for OC.

Line 294-299 – The authors state that they observed improved survival among patients who underwent TC during interval debulking surgery. How are they making that statement with no comparison group (i.e. patients who did not require colectomy or who underwent suboptimal or RD=1 cm, compared to RD0) debulking? Can they clarify this?

Line 304 – This is an important point that patient selection is important to obtain optimal outcomes from aggressive surgery while minimizing morbidity. Additional analyses as mentioned above to determine risk factors for poor outcomes (or factors associated with prolonged survival) could add to the discussion and would make this point stronger.

Author Response

Dear Reviewer,

We would like to thank you very much for your time and efforts in reviewing our work. We have corrected the manuscript accordingly to your remarks. Please find our responses below:

Question: This study describes oncologic outcomes and surgical morbidity among patients undergoing total colectomy for ovarian cancer debulking. This is an interesting and timely question, as the use of radical surgery to achieve maximal cytoreduction at the time of primary debulking in ovarian cancer has been questioned by several recent studies (i.e. SCORPION trial, Chorus trial, Vergote, etc.). As the authors point out, rare procedures such as colectomy are difficult to evaluate in randomized  trials, so a retrospective review is appropriate for this question.

Significance: This manuscript is significant, since it is the largest series of patients undergoing total colectomy for ovarian cancer debulking. However, I think some of their conclusions are overstated since there is no comparison group in this descriptive study.

Quality of presentation: The article is written appropriately, and the authors describe the questions and conclusions well. There are some areas where the English level is not that of a native speaker, but overall it is well-written.

Scientific Soundness: The study design is appropriate for the question. This is a small study, but the analyses were performed appropriately. The data is robust enough to draw some conclusions regarding the feasibility of TC in OC debulking. I think some additional analyses could also be considered.

Answer: We have performed two multiple regression analyses to indentify factors related with worse prognosis and surgically-related adverse evnets.

Question:

Interest to readers: This should be of interest to readers given the recent questions surrounding the necessity of performing high morbidity procedures for upfront debulking (vs. neoadjuvant chemotherapy with a lower complexity interval debulking surgery).

Overall merit: I think there is an overall benefit to publishing this work, though some additional analyses would improve the merit of this manuscript.

English level: The English is appropriate and understandable, though I think some minor edits would improve the readability of the manuscript.

Comments:

Line 137-139: I’m not sure Sugarbaker’s completeness of cytoreduction score is the best scale to use for this manuscript. I think using residual disease (RD) 0 vs. </= 1 cm vs. > 1 cm would be a more clinically relevant surgical outcome. Additionally, the difference between CC0 and CC1 is very minimal, so that may explain why there is no difference in outcome between these groups. Using the more clinically relevant outcomes of RD0, optimal debulking, and suboptimal debulking could yield different results.

Answer: You are absolutely right. However, we have used to Sugarbaker’s scoring system, and because of the retrospective character of the study, it is impossible to classify resudial disease in the three-stage system (R0, “optimal” debulking, suboptimal debulking). We have risen this issue in the discussion section.

Question: Line 156-165: The authors could report some additional clinical variables including nutrition status (albumin), estimated blood loss (EBL), disease stage, presence of ascites, AGO score, or other factors to more completely describe this group of patients. Could also do some additional analyses to determine whether specific patient characteristics such as nutrition status, age, length of surgery, EBL, ascites, etc. is associated with survival or morbidity in patients undergoing TC for OC.

 Answer: we were able to evaluate the impact of patient age, diaphragmatic stripping, splenectomy, liver metastasectomy, residual disease, BMI, previous chemotherapy, preoperative albumin level on presence of adverse events and patient prognosis.

Question: Line 294-299 – The authors state that they observed improved survival among patients who underwent TC during interval debulking surgery. How are they making that statement with no comparison group (i.e. patients who did not require colectomy or who underwent suboptimal or RD=1 cm, compared to RD0) debulking? Can they clarify this?

Answer: And again, you are right. We have modified this issue. We have changed, the „improved” for „promising” survival.

Question: Line 304 – This is an important point that patient selection is important to obtain optimal outcomes from aggressive surgery while minimizing morbidity. Additional analyses as mentioned above to determine risk factors for poor outcomes (or factors associated with prolonged survival) could add to the discussion and would make this point stronger.

Answer: We have performed additional analyses to identify factors related with patient survival and adverse events.

Round 2

Reviewer 1 Report

The authors have improved the manuscript to the better, however there are still some major concerns primarily to the methods and results section. I recommend that a statistician help them with improving these sections according to the suggestions below.

MAJOR:

Line 95: The authors confirm in their response that the study does not have a control group. For this reason, it may not be a cohort study. This is an observational and descriptive study. Please change.

Line 115 the sentence starting with “we present…” should be deleted.

Line 131 and 118: Suggest Figure 2 to be deleted. A flowchart Figure 1, on how the colectomy patients were selected should be provided in the results section including number of patients from the beginning and how they were deselected with reason to arrive to n=83 (see other papers for inspiration on flow charts of patient selection)

Section 2.4: Only list the confounding variables. The explanatory variable of interest is total colectomy the rest are confounders. Adverse events is an outcome not an explanatory variable or is it an explanatory variable used to adjust for the other outcome survival?

How the statistical analysis has been made is moved to 2.5 statistical analysis.

Section 2.5: Line 198: information on vital status and date of when this information was retrieved should be moved to the outcome 2.3 section.  Median follow up period need to be presented in the results section.

Define how you deselected variables stepwise in you regression how many variables did you enter from the beginning and how many did you use in the final model, or were the confounding variables predefined?

Define which statistical software you used for your analysis.

Results section:

Table 1: Include all variables you choose to adjust for in the multivariable regression for adverse events and cox regression.  The current Table 1 does not comply with the general characteristics.

Line 243: The Odds Ratio with Confidence intervals should be provided with information on the adjusted results.  Provide a table with both unadjusted and adjusted column with results of your outcome adverse events Clavien Dindo ≥ 3, including Odds Ratio, Confidence Intervals and P-values.

Since OS is your main outcome. Median OS with 95% CI on all women subjected to colectomy should be provided

In Figure 2 the Kaplan Meier curve define number at risk for each group and time (0, 20, 40, 60, 80 months)

Line 281: You state that you have used cox proportional hazards method in the statistical methods section therefore RR (relative risk) should be changed to Hazard Ratio and you should state if this is the results of the adjusted or unadjusted analysis. It is furthermore not clear if this Hazard Ratio calculated between which groups (PDS, NACT-IDS?) since it is under the section 3.2 Primary debulking surgery.